# Fumonisin Production by *Fusarium verticillioides* in Maize Genotypes Cultivated in Different Environments

**DOI:** 10.3390/toxins11040215

**Published:** 2019-04-10

**Authors:** Oelton Ferreira Rosa Junior, Mateus Sunti Dalcin, Vitor L. Nascimento, Fernando Machado Haesbaert, Talita Pereira de Souza Ferreira, Rodrigo Ribeiro Fidelis, Renato de Almeida Sarmento, Raimundo Wagner de Souza Aguiar, Eugenio Eduardo de Oliveira, Gil Rodrigues dos Santos

**Affiliations:** 1Plant Production Department, Universidade Federal do Tocantins—Campus Gurupi, Gurupi, Tocantins 77402-970, Brazil; oelton.junior@gmail.com (O.F.R.J.); m2d@uft.edu.br (M.S.D.); vitor.nascimento@uft.edu.br (V.L.N.); fernandomh@uft.edu.br (F.M.H.); rrfidelis@uft.edu.br (R.R.F.); rsarmento@uft.edu.br (R.d.A.S.); rwsa@uft.edu.br (R.W.d.S.A.); 2Biotechnology Department, Universidade Federal do Tocantins—Campus Gurupi, Gurupi, Tocantins 77402-970, Brazil; cupufer@gmail.com; 3Department of Entomology, Federal University of Viçosa, Viçosa, Minas Gerais 36570-000, Brazil; eugenio@ufv.br

**Keywords:** *Zea mays* L., mycotoxins, temperature, maize ear rot

## Abstract

Fumonisins are mycotoxins (MTs) produced mainly by the fungus *Fusarium verticillioides*, the main pathogens of maize which cause ear rot. The aim of this work was to evaluate some factors that may lead to high fumonisin production by *F. verticillioides* in maize grains, correlating the pathogen inoculation method with different genotypes grown in four Brazilian states. Experiments were conducted in 2015–2016 in maize crops from experimental maize fields located in four distinct states of Brazil. Results showed that contamination by fumonisin mycotoxins occurred even on symptomatic or asymptomatic grains. In all municipalities, the samples showed levels of fumonisin B1 that were higher than would be tolerable for the human consumption of corn products (the current tolerance limit for fumonisin is 1.5 μg g^−1^). High severity of grains infected with *F. verticillioides* does not always show high concentrations of fumonisins. Environments with higher temperatures may influence the production of high concentrations of fumonisin in maize hybrids. Spray inoculation methods and inoculation at the center of spikes did not influence fumonisin concentrations. Results showed that the hybrids P3630H, P32R48 and P3250 presented higher disease severity, as well as higher mycotoxin levels in the studied locations with higher temperatures.

## 1. Introduction

At present, Brazil is the third-largest producer of maize (*Zea mays* L.) in the world, generating more than 92 million tons over the 2016–2017 harvest season, with an average productivity of 5.409 kg per hectare [1]. Among the problems of this crop are diseases that diminish productivity and cause large economic losses. The most important diseases associated with maize in Brazil are caused by fungi, which are primarily represented by the genus *Fusarium.* In addition to being able to attack the roots, leaves and stems of plants, this fungus causes rot in the ear and damages the grains, compromising yield; it is often identified only when the grain is delivered to processing centers [2]. Two Fusarium-induced ear rots have been reported in Brazil: Fusarium ear rot, caused by species within the *Gibberella fujikuroi* complex, especially its anamorphic species *F. verticillioides*, *F. subglutinans* and *F. proliferatum*; and Gibberella Ear Rot, caused by *Gibberella zeae*, the teleomorphic stage of the *F. graminearum* species complex [3,4]. In an exploratory survey carried out in corn-producing regions in the state of Rio Grande do Sul, Brazil, [4] it was shown that *F. verticillioides* was the dominant species (76%) of the isolates obtained. This species produces mycotoxins (MTs), especially fumonisin, which are frequently found in maize and derivatives [5]. These toxins are found in different regions and they predominate in tropical and subtropical climates, such as Brazil, where the development of several microorganisms adapted to the tropical climate is favored by high temperatures and humidity levels [6]. Climatic conditions can be a decisive factor in the production of mycotoxins [7].

The major classes of fumonisin are FB1, FB2 and FB3 [8,9]. To date, 28 analogous compounds have been identified as fumonisins, with fumonisin B1 (FB1) being the compound with the highest toxigenicity that is most frequently found in food. These compounds are associated with approximately 70% of natural fumonisin contamination [10]. These toxins cause leukoencephalomalacia in horses and rabbits [11], swelling and pulmonary edema in pigs [12] and hepatic cancer in rodents [13]. The effects of human exposure to fumonisins from diet remain uncertain. However, FB1 intake has been correlated with the incidence of esophageal cancer in several parts of the world such as China [14], Africa [15,16], Iran [17], Argentina and Brazil [18]. In addition, FB1 is classified by the International Agency for Research on Cancer (IARC) as belonging to group 2B (i.e., a possible carcinogen in humans) [19].

Due to the problems caused by this class of mycotoxin, international agencies have established maximum fumonisin concentrations in maize for food and feed production. For instance, depending on the food product, the US FDA (United States Food and Drug Administration) recommends maximum levels between 2 and 4 μg g^−1^, and the EFSA (European Food Safety Authority) recommends maximum levels between 0.8 and 4 μg g^−1^ [20]. In Brazil, the Agência Nacional de Vigilância Sanitária-ANVISA (National Agency of Sanitary Surveillance) has established the following maximum limits: (*i*) maize grain, 5 μg g^−1^; (*ii*) corn flour, corn cream, corn meal, flakes, canjica and canjiquinha, 1.5 μg g^−1^; and (*iii*) maize starch and other maize-based products, 1 μg g^−1^ [21]. These pre-established limits are of great importance because they are substances of great chemical and thermal stability that may persist in food or feed even after chemical decontamination through industrialization processes. When metabolized by animals, they can occur in meat, eggs and milk, posing a risk to human health [21,22].

In regions with a tropical climate (e.g., Brazil, Argentina and South Africa), there is a lack of research aimed at identifying maize genotypes resistant to *F. verticillioides* infections and fumonisin accumulation, culminating in a limited number of genotypes resistant to both infections with the pathogen and accumulation of the mycotoxin [23,24,25]. In Brazil, as well as in some countries on the Asian and African continents, there are environmental factors, mainly high temperatures and relative humidity, favoring the development of ear rot and its pathogen *F. verticillioides*. Specifically in Brazil, corn is grown in regions with different soil types, climate, genotypes (varieties and hybrids) and in different cropping systems. Thus, it is believed that under these conditions there is a greater inoculum potential of the pathogens, and variability in the durable resistance of the plants can also be verified. Factors that make research results more difficult to obtain can be attributed to the complexity for conducting trials in different locations and the high cost of chemical analysis. Normally, analysis of mycotoxins (MTs) requires extraction procedures and very specific analysis to enable careful assessment of the risk in different foods, and to estimate the average daily or weekly intake of MTs [26]. In Brazil, the contamination of food by fumonisin has been addressed by several authors, especially in maize and its derivatives [25,27]. However, little is known about the production dynamics of these mycotoxins by *F. verticillioides*, and there are no studies to date that correlate these fumonisin levels with other factors.

The lack of efficient mycotoxin monitoring indices in grains allows high concentrations of fumonisin to pass through the system unnoticed and be processed along with the grains. In addition, *Fusarium* is a microorganism that may be endophytic [28,29] or cause latent infections. Thus, little is known about the relationship between the presence of this pathogen in maize kernels and its predicted rates of fumonisins. In this way, it is possible to adopt production practices that decrease the levels of this mycotoxin in maize kernels. Therefore, the objective of this work is to evaluate the effects of environmental factors, genotypes and inoculation methods pertaining to *F. verticillioides* on fumonisin production in maize grains.

## 2. Results and Discussion

The present work evaluates the effects of inoculated *F. verticillioides* on different corn genotypes grown in four Brazilian regions, as well as evaluating the severity of corn ear rot disease, fumonisin content and grain yield.

### 2.1. Analysis of Variance

According to the analysis of variance (Appendix A), a significant interaction was observed between the treatments and environments; thus, an analysis was performed in each environment separately. The interaction between the inoculation and the genotype was not significant for disease severity and thousand kernel weight (TKW), and thus no specific analyses were performed when considering the separation of each of the inoculations. Although the interaction between the genotypes and inoculation was not significant, the isolated effect of each of these factors was significant. That is, according to the analysis of variance, there was no correlation between inoculation and genotypes. Thus, when the inoculation method changed for each genotype, there was no variation in the TKW response pattern and disease severity; however, the effect of the genotypes was significant for these variables, as well as for inoculation methods.

### 2.2. Identification of Isolates

Figure 1 shows the molecular weight of DNA samples from the fungal isolates obtained from ears containing symptomatic seeds. A sample resulted in the amplification of a 658-nucleotide product for the region EF, and this sequence showed 99% shared identity with *F. verticillioides* (accession # FN179338.1), covering 99% of the sequence according to a BLAST/NCBI query. In relation to the morphological analysis of the isolate, microscopic structures typical of the pathogen were also visualized (Figure 1) in which conidiophores with microconidia formed in long chains, and monophialides were observed. Based on the results obtained in the molecular analysis, as well as in the morphological characteristics, the evaluated isolate was *F. verticillioides* (Sacc.) Nirenberg.

### 2.3. Production of Fumonisins Depending on the Location and Inoculation Methods

The presence of rainfall in all localities where the trials were conducted created favorable conditions for the development of corn ear rot in the evaluated genotypes.

Regarding the natural infection of the hybrids, there were high levels of fumonisins in three of the four municipalities evaluated here, but in all the inoculated locations high levels of fumonisin were detected (Table 1). Considering that there was no artificial inoculation with an infection in the treatment, it is worrying to see very high indices due to natural infection.

Approximately 70% of samples from the municipality of Gurupi, TO presented fumonisin B1 (FB1) and 40% of the samples had levels of fumonisin B2 (FB2) that were considered higher than tolerable if intended for human consumption (in corn flour, corn cream, corn meal, flakes, canjica and canjiquinha), with 1.5 μg g^−1^ being the current tolerance limit for fumonisin. Similarly, if the intended use was maize starch production, approximately 90% of the samples would be above the tolerance level for FB1 and 50% would be above the tolerance level for FB2, because in this case, the fumonisin limit is 1 μg g^−1^. The mean severity of the controls of this experiment was 3.44, in which this score represents 18.8% of the ear area that exhibited visible symptoms of fungal infection. Under these conditions, the concentrations of FB1 were 0.4 μg g^−1^ to 16.2 μg g^−1^, considering an average of 5.48 μg g^−1^, which is almost four times higher than the limit officially established by ANVISA. Similar results were also found by Hirooka et al. [30], who analyzed 48 maize samples from Paraná, Mato Grosso and Goiás and detected fumonisins B1 and B2 in all samples, with levels ranging from 0.6 to 18.52 μg g^−1^ FB1 and from 1.2 to 19.13 μg g^−1^ FB2. Yoshizawa et al. [8] analyzed 18 samples of Thai maize from the 1992–1993 crop, detecting FB1 in 89% of the samples (0.063–18.8 μg g^−1^) and FB2 in 67% (0.05–1.40 μg g^−1^).

According to the analysis of mycotoxin FB1 in grains from Itumbiara, GO, 90% of the control samples had levels above the tolerated limit. For this reason, these grains would be inappropriate to use for the manufacture of corn flour, corn cream, corn meal, flakes, canjica or canjiquinha, as well as for maize starch and other maize products. Almeida et al. [31] verified FB1 and FB2 contamination in 57 samples of maize grains from the Capão Bonito and Ribeirão Preto regions at the different maturity stages of the plant. The results from Capão Bonito showed FB1 contamination in 92.3% of the samples (50–10.87 μg g^−1^), and FB2 contamination in 61.5% (0.05–0.521 μg g^−1^). Samples from the Ribeirão Preto region showed that 96.8% of the samples were contaminated with FB1 (50–17.69 μg g^−1^) and 74.2% with FB2 (0.05–5.24 μg g^−1^).

In grain produced in Planaltina, DF, FB1 analysis revealed that 50% of the control samples presented levels above the tolerated limit, and 10% of the samples would also be above this same tolerance level for FB2 if the grains were processed for the same derivatives described above. Orsi et al. [32] studied 195 samples from three hybrids of freshly harvested and stored maize from Ribeirão Preto, São Paulo state, and found that 90.2% of the samples were contaminated with FB1 (0.87–49.3 μg g^−1^) and 97.4% with FB2 (1.96–29.16 μg g^−1^).

For grain analysis from Toledo, PR, high FB1 levels were verified only in the inoculated genotypes. In this municipality, 70% of the inoculated grain samples presented concentrations above the tolerated levels for use in corn flour, corn cream, corn meal, flakes, canjica and canjiquinha, and 80% would be above the tolerated limit for maize starch. In the control, in which the ears were not artificially inoculated, no fumonisin FB2 analog was found and only two hybrids showed fumonisin at the same concentration of 0.27 μg g^−1^ FB1. The low levels may have been influenced by the milder climate, according to Table 2 and Figure 2.

Table 2 shows the fumonisin concentrations in the inoculated genotypes of each location as well as the temperature averages. The highest FB1 concentration occurred in the municipality of Gurupi, at 24.84 μg g^−1^, which is 24 times above the tolerance limit if the grains were intended for use in maize starch or other maize products. At this location, the average temperature was 27.43°C. The lowest concentration of FB1 occurred in Toledo, with a mean concentration of 3.38 μg g^−1^, or more than three times the tolerance limit if the grains were destined for use maize starch or other maize products. The average temperature at this location was 13.64°C. The results showed increasing levels of fumonisins, with the highest concentrations occurring in places that experienced higher temperatures. However, this occurred according to the average of the cultivars added in each region (Table 2). Observing the amount of fumonisin for each genotype, those that showed a positive correlation with temperature were 30K75Y, 32R48YH, DKPDKB240PRO2, DKPDKB390PRO2, P3250, P3340YHR and P3630H (Appendix A).

In the present work, FB1 concentrations were three to four times higher than FB2 (Table 3). This finding was verified in practically all samples from the analyzed locations. In the general correlation analysis, a significant positive correlation (0.99) between FB1 and FB2 was obtained, demonstrating that despite the difference in concentration, these types of fumonisins are produced in association with one another, so that when one increases, the other one does as well.

### 2.4. Influence of Genotypes on Grain Weight, Disease Severity and Fumonisin Concentrations

In the municipality of Gurupi, TO, the average minimum temperature was 19.9°C and the maximum was 34.9°C during the experimental period (Figure 2). In this environment, as shown in Table 3, the DKB310PRO2 hybrid presented the lowest level of severity, at 2.56 (5.82% of the ear area showed visible symptoms of fungal infection), as well as the largest thousand kernel weight (338 g) and one of the lowest levels of fumonisin FB1 (2.69 μg g^−1^) in the experiment.

Hybrid correlation analyses (Appendix A) showed a negative correlation between the TKW and a 0.70 disease severity for the DKB310PRO2 hybrid (a higher TKW indicates lower severity). There was a positive correlation of 0.75 between severity and the concentrations of fumonisins FB1 and FB2. Thus, low levels of disease severity resulted in low fumonisin values for this hybrid. Similar results were found by Afolabi et al. [23], who verified that lower levels of severity result in low fumonisin values.

The 32R48YH hybrid presented high disease severity, at 6.56 (75.6% of the ear area exhibited visible symptoms of fungal infection), and a high TKW (319.89 g). In this hybrid, fumonisin content was 64.8 μg g^−1^, which amounts to almost 65 times the tolerated limit for some maize derivatives [21]. This hybrid showed the highest disease severity and the second-highest fumonisin content of the plants studied. In the correlation analysis (Appendix A), a value of 0.73 was obtained (a higher severity indicates a higher mycotoxin content). Both analogs FB1 and FB2 presented a positive correlation with high temperatures. Thus, as the temperature increased, the concentration of mycotoxins increased, with 0.70 for FB1 and 0.66 for FB2. Similar results were found by Afolabi et al. [23], who reported that the fumonisin concentration showed a significant correlation with the severity of ear rot and incidence of symptomatic grains. The same finding occurred in studies conducted by Desjardins et al. [12] and Kleinschmidt et al. [33] in Nigeria and different American states.

The DOW30A37PW hybrid presented one of the highest severities and the lowest TKW (254.56 g) of the experiment, which may have been influenced by the infection in the ears, consequently resulting in a lower grain weight. There was a negative correlation of −0.6%, and a higher severity resulted in a lower TKW. In this hybrid, the lowest fumonisin B1 content (1.68 μg g^−1^) of the experiment was found, and it may be hypothesized that the fumonisin levels could have been influenced by the genotype, because, for this hybrid, the relatively high severity did not express high levels of fumonisin as observed in the 32R48YH hybrid. According to Sánchez-Rangel et al. [34], these results may be related to the following two hypotheses: (*i*) some of the symptoms caused in the ear were caused by isolates other than those of *F. verticillioides* that were not producing fumonisins, or that were producing levels below the detectable limit; and (*ii*) a low relation between disease intensity and levels of fumonisins is an inherent factor of this pathosystem. This latter hypothesis is reinforced by other studies, such as Afolabi et al. [23], suggesting that genetic factors that affect grain infection may act independently of those affecting fumonisin production, as in Munkvold and Desjardins [15], which is partly explained by the quantitative nature of these two characteristics.

The hybrid P3630H showed low to intermediate severity, at 3.78 (25.56% of the ear area exhibited visible symptoms of fungal infection). It had a thousand kernel weight of 295.22 g and the highest FB1 content found in Gurupi, TO, with a content of 86.7 μg g^−1^. There was a positive correlation between the high concentrations of fumonisin in this hybrid and high temperatures (Appendix A). The low severity of the disease and the production of high fumonisin levels have also been verified by Saunders et al. [35], who noted that there is generally no difference in the visual appearance of maize with low or high concentrations of fumonisins in relation to severity levels. On the other hand, we can compare it to another hybrid such as 32R48YH, in which high severity of disease in the ears correlated with high levels of the mycotoxin (a correlation of 0.73). The P3630H hybrid demonstrated that, in addition to severity level, other variables may be related to the production of mycotoxins, including genetics.

Hybrids P3630H and 32R48YH had the highest fumonisin indices at Gurupi-TO (Figure 3). A possible effect of the genetics of each genotype was also verified in hybrid P3250, where it was observed that even with medium-severity values, notable fumonisin levels were found that were more than 30 times the tolerance limit determined by ANVISA [21]. Thus, this result leads us to believe that fumonisin content is not always associated with high ear rot severity.

In the present work, hybrids such as P3630H and P3250 had medium levels of disease severity in the ears, a similar TKW and high levels of fumonisins. However, the DOW30A37PW hybrid experienced medium to high disease severity (the second-highest severity of the experiment) and presented very low rates of FB1 (the lowest FB1 content of the experiment). These results again demonstrate that high severity may not result in high mycotoxin indices, but could be an intrinsic characteristic of each hybrid, highlighting the influence of the genetics of each genotype. In this sense, Eller et al. [36] and Butrón et al. [20] report that efforts should be directed at finding genetic material that is resistant to both parameters, because there is not always a correlation between grain infection and fumonisin levels. Furthermore, as described by Munhoz et al. [37], rains without ear rot are not guaranteed to show an absence of fumonisin contamination or fumonisin levels below the maximum tolerance limits. According to a number of authors [12,23,38], there are at least three possible interactions in this model between the symptoms and the toxins, as follows: (*i*) the presence of symptoms in the grains accompanied by high levels of fumonisins; (*ii*) the presence of symptoms and low fumonisin contents; and (*iii*) the production of toxins in grains with no visible signs of infection.

In the municipality of Itumbiara, GO, an average minimum temperature of 18.30°C and a maximum of 27.85°C (Figure 2) were recorded during the experimental period. In this environment, the 4285H hybrid presented the lowest severity of the disease and the largest TKW, at 279.89 g, with the second-lowest concentration of fumonisin at 0.568 μg g^−1^ (Table 3). The DKB310PRO2 hybrid showed intermediate disease severity, the second-largest TKW (272 g) and an intermediate fumonisin concentration (8.67 μg g^−1^) when compared to the concentrations of the other hybrids. However, this figure represents approximately nine times the tolerance limit determined by ANVISA [21] for grains used in maize starch.

The P3250 hybrid showed high disease severity at 5.33 (56.5% of the ear area exhibited visible symptoms of fungal infection), an intermediate TKW of 217.44 g and the highest concentration of fumonisin in the experiment, which corresponded to 18.81 μg g^−1^ (Table 3). Similar results were found for the 30K75Y hybrid, which also presented high disease severity at 6.11 (71.1% of the ear area exhibited visible symptoms of fungal infection) and the second-highest level of fumonisin in the experiment (12 μg g^−1^). Studies developed by Eller et al. [36] in different American states revealed that high fumonisin concentrations correlated with the presence of symptomatic grains and a high severity of ear rot. The hybrids P3250H and 30K75 displayed the highest rates of fumonisin at the evaluated location (Table 3).

Hybrid DKB240PRO2 (Table 3) showed a high disease severity of 6.11 (71.1% of the ear area showed visible symptoms of fungal infection), a low TKW (159.89 g) and an FB1 concentration that was almost six times higher than the tolerance limit determined by ANVISA [21] (5.66 μg g^−1^). As shown in Appendix A, there was a negative correlation (−0.81) in the analysis of disease severity and grain weight, and thus, a higher severity resulted in a lower TKW.

The P3340YHR hybrid presented the highest severity of the experiment, with an intermediate grain weight and a fumonisin concentration of 8.04 μg g^−1^, corresponding to eight times the tolerance limit [20]. Although the severity of the disease in this hybrid was higher (6.22) than it was in the P3250 hybrid, the highest concentrations of mycotoxin were found in the latter—8.04 μg g^−1^ and 18.81 μg g^−1^, respectively. Once again, these findings showed that high disease severities do not always indicate the highest fumonisin concentrations in maize hybrids.

In the municipality of Planaltina, DF, the average minimum temperature was 17.72°C and the maximum was 26.54°C (Figure 2) during the experimental period. In this environment, the 32R48H hybrid had the highest level of disease severity, at 5.33 (56.6% of the ear area exhibited visible symptoms of fungal infection), as well as the highest fumonisin content (11.07 μg g^−1^) and the largest TKW (381.78 g). A correlation of 0.73 between disease severity and fumonisin concentration was verified for this hybrid (Appendix A). Both analogs FB1 and FB2 also presented a positive correlation with high temperatures. Thus, for this hybrid, as the temperature increased, the mycotoxin concentration grew, as evidenced by values: 0.70 for FB1 and 0.66 for FB2. The similar performance of this hybrid was verified in the Gurupi, TO experiment, in which relatively high disease severity and high fumonisin indices were associated. The 32R48YH hybrid exhibited the highest fumonisin index at the location, which can be best observed in Figure 3.

The P3250 hybrid presented a low disease severity of 3.44 (18.8% of the ear area exhibited visible symptoms of fungal infection) and the lowest concentration of fumonisin in the experiment (1.68 μg g^−1^). For this genotype, low severity reflected a low fumonisin content at this location. A similar result was found by Afolabi et al. [39] during studies in Nigeria, and they reported that the fumonisin concentrations showed a significant correlation with the severity of ear rot and with the incidence of symptomatic grains.

The P3630H hybrid in Planaltina, DF also exhibited results similar to those found in Gurupi, TO. It showed an intermediate disease severity and high levels of fumonisins, with the second-highest content of this mycotoxin (6.95 μg g^−1^). These values were almost seven times above the fumonisin tolerance limit [21] for grains destined for the production of maize starch. In correlating disease severity with fumonisin concentration, we can observe a positive correlation equal to 0.60 (Appendix A); that is, at this location, for this hybrid, there was a tendency for high mycotoxin concentrations to be associated with high disease severity.

In the municipality of Toledo, PR, the average minimum temperature was 9.03°C, and the maximum temperature was 18.25°C, during the experimental period (Figure 2). In this municipality, the P4285H, DOW30A37PW and P3630H hybrids presented similar behaviors, with medium disease severity followed by the highest concentrations of fumonisin at the location—4.56 μg g^−1^, 6.87 μg g^−1^ and 7.57 μg g^−1^, respectively. These values corresponded to 4.5–7.5 times the tolerance limit determined by ANVISA [21] for grains intended for use in maize starch production. Low disease severity and few visible symptoms were suggested in the work of Afolabi et al. [23], Desjardins et al. [12] and Picot et al. [38], who reported few visible symptoms in the tested genotypes, but high levels of fumonisin. Hybrids P4285H, DOW30A37PW and P3630H had the highest fumonisin indices at the location, which can be observed in Figure 3.

The P3630H hybrid exhibited the same result found in the municipalities of Gurupi, TO and Planaltina, DF. In Toledo, PR, the disease severity was once again low to intermediate, at 3.0 (10% of the ear area exhibited visible symptoms of fungal infection), and showed the highest fumonisin content at that location (7.57 μg g^−1^).

In contrast to the results obtained at the other locations, the 32R48YH hybrid presented a very low concentration of fumonisin B1 (0.2 μg g^−1^), which might have been influenced by climate (there was a 13.64°C temperature during the experiment). The correlation between the fumonisin level and the temperature was positive, at 0.70. This factor likely influenced the low concentration of mycotoxin in this hybrid. Similar situations were reported by Herrera et al. [40] and Munkvold and Desjardins [15], in which environmental conditions such as low temperatures and relative humidity are suggested to contribute to lower disease development, and possibly result in a lower accumulation of fumonisins. In another paper, Blandino et al. [41] argue that low temperatures reduce insect attacks, leading to a reduction in *F. verticillioides* infection and levels of fumonisins.

Results showed that the hybrids P4285, DKB390PRO2 and DKB310PRO2 were the most resistant in the majority of locations studied, while the hybrid 32R48YH showed the highest severity of fumonisin. The hybrids P3630H, P32R48 and P3250 presented higher disease severity, as well as higher mycotoxin levels in the locations with higher temperatures (Figure 2 and Figure 3).

In Brazil, this is the first study to relate the severity of the disease and fumonisin levels in maize hybrids through the inoculation of potentially toxigenic isolates, as tested in four different states of Brazil.

## 3. Conclusions

In all municipalities the samples showed levels of fumonisin B1 that were considered to be higher than tolerable if destined for human consumption in corn products, with the tolerance limit for fumonisin currently being 1.5 μg g^−1^.

Contamination of grains by fumonisin mycotoxin occurs even in symptomatic or asymptomatic grains. The spray inoculation methods and inoculation at the center of the ear did not influence the different concentrations of fumonisins.

Generally, in some genotypes, the production of fumonisins in maize grains can be potentiated by high-temperature environmental conditions and by the susceptibility of the genotype. The severity of ear rot maize was associated with higher fumonisin contents for most of the genotypes evaluated here.

Maize hybrids P4285, DKB390PRO2 and DKB310PRO2 were the most resistant to ear rot maize in the four environments, while the hybrids P3630H, 32R48 and P3250 presented the highest severity of the disease. Hybrids P3630H, 32R48 and P3250 presented in most locations with the highest concentrations of fumonisins.

Grain productivity is reduced by the incidence of *F. verticillioides.*

## 4. Materials and Methods

### 4.1. Experimental and Treatment Locations

These experiments were conducted during the 2015–2016 harvest in experimental maize farms located in distinct areas of Brazil, including the municipalities of Gurupi, Tocantins state (TO); Toledo, Paraná state (PR); Planaltina, Distrito Federal (DF); and Itumbiara, Goiás state (GO), as shown in Figure 4.

Gurupi is located in the Cerrado biome and, according to the Koppen and Geiger classification [42], the climate of the region is Aw, which is defined as tropical hot and humid with a rainy season in the summer and a dry winter (the altitude is 278 m). The period for installing and conducting the experiment was from February to June of 2016.

Toledo is located in the Atlantic Forest biome at an altitude of 550 m. The climate of the region is Cfa, which is defined as subtropical, with hot summers, infrequent frosts and no defined dry season. The period for installing and conducting this experiment ran from February to July of 2016.

Planaltina is located in the Cerrado biome and has an altitude of 1175 m. The climate of the region is Aw, defined as tropical hot and humid, with a rainy season in the summer and a dry winter. The period for installing and conducting this experiment was from December 2015 to April 2016.

Itumbiara is located in the Cerrado Biome and has an altitude of 448 m; the climate of the region is Aw, which is tropical, hot and humid, with the rainy season in the summer. This experiment was installed and conducted from December 2015 to April 2016.

At the four experimental locations, the implementation of a meteorological mini-station capable of recording the climatic variables and providing a microclimate favorable to disease development in the ears was performed, starting from inoculation, and running from flowering to harvesting. The climatic variables consisting of the maximum and minimum temperatures (°C) and precipitation (mm) at the four locations were monitored and recorded from the inoculation of the pathogen to the grain harvest (Figure 2).

The experimental design was the same in the four studied localities, and consisted of a randomized block design in a 10 × 3 factorial scheme with 10 maize genotypes, 3 inoculation methods, and 3 replicates. The 10 maize hybrids used here were P30K75Y, P32R48YH, P3250, P3340YHR, P3630H, P4285H, DKB240PRO2, DKB390PRO2, DKB310PRO2 and DOW30A37PW. The hybrids were chosen due to their large commercial cultivation in several Brazilian regions.

To inoculate *F. verticillioides* into the ears, a suspension of the fungus was obtained from its cultivation in BDA culture medium. Isolates of the species from their own regions were used in each location, as they are commonly more adapted to the climate of the study region. The suspension inoculation was standardized to 5 × 10^5^ conidia mL^−1^.

The three methods of inoculation were as follows: (*i*) injecting the conidial suspension into the center of the ear and simulating the fungal infection through physical damages caused, for the most part, by insects; (*ii*) spraying a suspension of conidia in the form of a spray under the style-stigmas and simulating the natural spread of the spores reaching the stigmas of the ears; and (*iii*) the natural method, in which there was no artificial inoculation (this was the control). After inoculation, a paper package was used to cover each ear to provide a better environment for the development of the pathogen and serve as a barrier against interference from other plots.

Each replicate consisted of six rows of 4.5 meters with 30 plants per row. Each inoculation method was applied to two rows. In the treatments receiving inoculations, all of the plants in the rows were inoculated except the first two and last two. For both artificial methods, a 2-mL inoculation of the suspension into each ear was performed when the plants were in phenological stage R1 (flowering and pollination), a period of greater susceptibility to infection [15]. Two rows were assigned to the control treatment.

### 4.2. Isolation of the Pathogen, Koch’s Postulates and Molecular Identification

To isolate the pathogen, symptomatic seeds from each location were analyzed for the presence of *F. verticillioides*. The fungus was transplanted into Petri dishes containing BDA culture medium (200 g of potato, 20 g of dextrose and 20 g of agar in 1 L of distilled water). The isolates were identified according to their typical morphology, which was characterized by the presence of long chains of microconidia produced in monophialides [43]. To purify the colonies from the isolates, a monosporic culture technique [44] was used in which a suspension of the fungal conidia was placed in 5 mL of distilled water and applied to sterilized Petri dishes containing agar-agar medium (AA: 20 g of agar in 1 L of distilled water). After 24 hours of incubation at room temperature, the germinated conidia were observed under an optical microscope. The conidia that germinated were individually transferred into test tubes and Petri dishes containing BDA culture medium for further inoculation work. Koch’s postulates were tested by applying inoculations of the fungal suspension containing 5 × 10^5^ conidia mL^−1^ to healthy 32R48YH hybrid plants. The inoculations were performed two areas, one in the style-stigmas and the other in the center of the ear. After 25 days of inoculation, when the plants were in phenological stage R1, all the ears were harvested to identify the pathogens present in the grains.

DNA extraction was performed on the fungal isolates grown in BDA medium according to the protocol described by Doyle [45]. A total of eight samples were used for the molecular analysis, with two origins per location in the municipalities of Gurupi, Itumbiara, Planaltina and Toledo. Approximately 1 mg of *Fusarium* mycelium was homogenized in a 2-mL microcentrifuge tube with the help of a pestle, using 600 μL of the 2% CTAB extraction buffer (1.4 M NaCl, 20 mM EDTA, HCl pH 8 100 mM and 1% polyvinylpyrrolidone) and 0.2% β-mercaptoethanol. After homogenization, the suspension was incubated at 65 °C for 30 min. Then, 50 μL of chloroform and isoamyl alcohol (24:1) were added, and the mixture was centrifuged at 14,000 RPM for 10 min. The supernatant was transferred to a new 1.5-mL tube, and 500 μL of chloroform and isoamyl alcohol was added. After that, centrifugation was performed at 14,000 RPM for 10 min. The supernatant was transferred to a new 1.5-mL tube and the same volume of ice-cold isopropanol was dispensed at −20 °C. The mixture was homogenized and maintained at −20 °C for 30 min, then centrifuged at 14,000 RPM for 15 min for DNA precipitation. The supernatant was discarded, and the precipitate was washed with 70% ethanol and diluted in 50 μL of milli-Q water. The DNA was stored at −20 °C until use in PCR reactions.

The PCR reaction used for the detection of *F. verticillioides* was performed with a mixture of 5.0 μL of the buffer (100 mM Tris HCl pH 8.5 and 500 mM KCl), 1.0 μL of dNTPs (0.2 mM), 1 μL of MgCl_2_, 2.5 μL of the sense and antisense primers (0.5 mM), 1.0 μL of TaqDNA polymerase (LGC) (5U μL^−1^) and 5.0 μL of the total DNA, with the final volume of the reaction adjusted with milli-Q water up to 50.0 μL. The primers VER1 and VER2 were described by Mulè et al. [46]. The thermocycler regimen for this reaction was 95 °C for 5 min, followed by 25 cycles of 1.5 min at 95 °C, 1 min at 56 °C, 1.5 min at 72 °C and a final extension step of 72 °C for 10 min. The product of this amplification was subjected to conventional electrophoresis in a 1% agarose gel and visualized in a UV transilluminator. Isolates from the same sample that was subjected to PCR analysis were sent to the Agronomic Laboratory for a second PCR analysis and sequencing to confirm the species.

### 4.3. Evaluation Infection Level of F. verticillioides

When the genotypes were in phenological phase R6, which is characterized as physiological maturation, grain moisture monitoring was initiated and maintained until the moisture reached 17%. A total of 150 ears were then collected, with 50 ears per inoculation treatment. The ears were subsequently arranged in the automated dryer so that the grains reached a humidity of 13%. The ears were evaluated for fungal severity, then threshed and analyzed for their grain weights and levels of mycotoxin fumonisins B1 and B2.

The evaluation of the severity of the disease was performed visually and comparatively, in which a score was assigned according to the different aspects of the ears. Severity was evaluated by means of a diagrammatic scale that estimates the percentage of the ear area with rot symptoms, characterized by a covering of white to pink mycelium, or the presence of dark grains and/or white streaks in the pericarp. As previously described by Agroceres [47], the scale grades the area of the ear exhibiting visible symptoms of infection: 1 = 0% (healthy), 2 = 0.5%, 3 = 10%, 4 = 30%, 5 = 50%, 6 = 70%, 7 = 80%, 8 = 90% and 9 = 100%. The hybrids that received grades of 1–3 were classified as having high to medium resistance, those that received scores of 4–6 were classified as having medium susceptibility and the hybrids that received scores from 7–9 were classified as having medium to high susceptibility. The evaluation of the thousand kernel weight, in grams, was performed right after the threshing of the ears, by counting one thousand grains from each treatment line for all the replicates and weighing them in a calibrated scale. The quantification of fumonisin (FB1 and FB2) was performed at the Laboratory of Mycotoxology (LAMIC), Federal University of Santa Maria (UFSM), with a Liquid Chromatography Mass Spectrometer (LC-MS/MS) POP 45.

### 4.4. Statistical Analyses

Before the analysis of variance, the data were tested with respect to the assumptions given by the mathematical model of error normality by a Shapiro–Wilk test, and the homogeneity of error variance was tested using a Bartlett test. When the data did not meet one of these assumptions, they were transformed by a Box-Cox method. The data were subjected to an analysis of variance for each location, a subsequent analysis of variance for all locations and finally a Tukey’s test for a comparison of means for the given factors, and to reveal significant interactions. The fumonisin data were correlated with the values for the thousand kernel weight, severity and maximum and minimum temperatures.

## Figures and Tables

**Figure 1 toxins-11-00215-f001:**
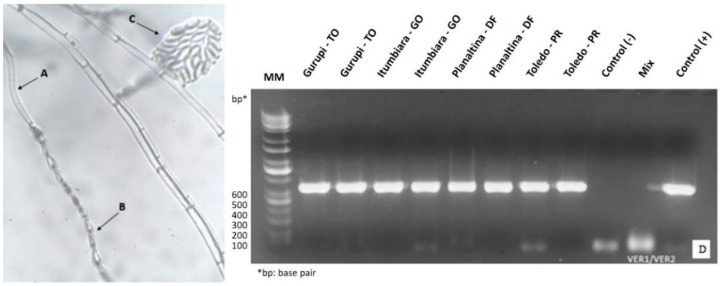
Morphological structures of *Fusarium verticillioides*, being, (**A**) conidiophore; (**B**) microconidia in a long chain; (**C**) microconidia in false heads; (**D**) molecular weight of samples of fungal isolates obtained from maize, cultivated in different locations: Gurupi, Tocantins (TO); Itumbiara, Goiás (GO); Planaltina, Federal District, (DF) and Toledo, Paraná, (PR). Negative control (Control −): sound corn leaf; Mix: relative control, no DNA addition; Positive control (Control +): isolated from *F. verticillioides*.

**Figure 2 toxins-11-00215-f002:**
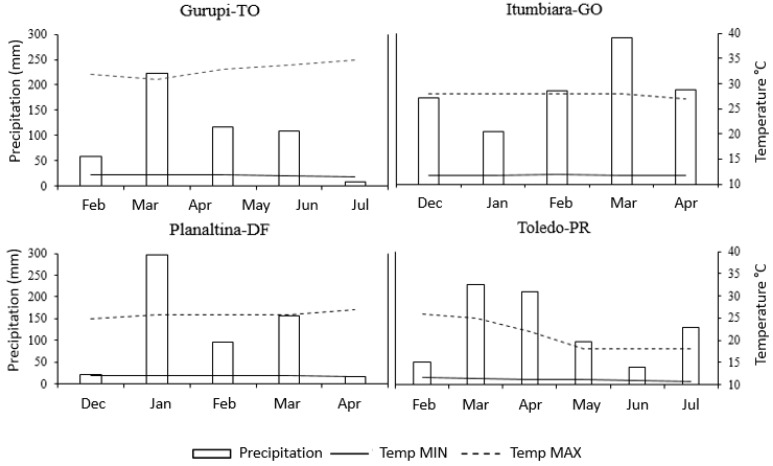
Climatic variables represented by the maximum and minimum temperatures (°C) and precipitation (mm) verified in Gurupi, Tocantins (TO); Itumbiara, Goiás (GO); Planaltina, Federal District (DF); and Toledo, Paraná (PR) during the conduction of the maize experiment.

**Figure 3 toxins-11-00215-f003:**
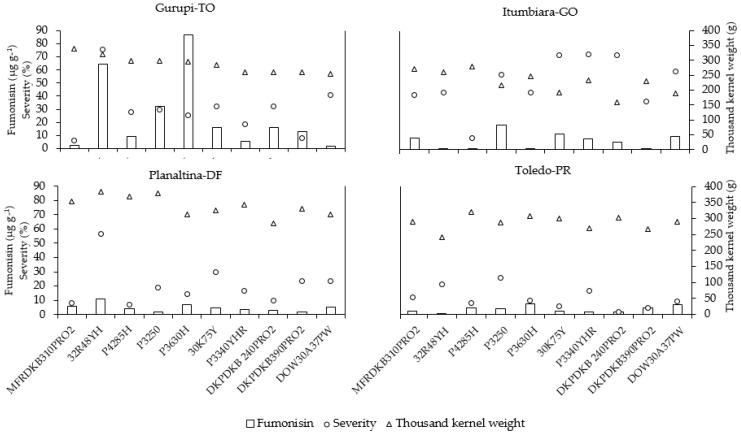
Values of thousand kernel weight, severity and concentration of fumonisin of maize hybrids in the municipality of Gurupi, Tocantins (TO); Itumbiara, Goiás (GO); Planaltina, Federal District (DF); and Toledo, Paraná (PR).

**Figure 4 toxins-11-00215-f004:**
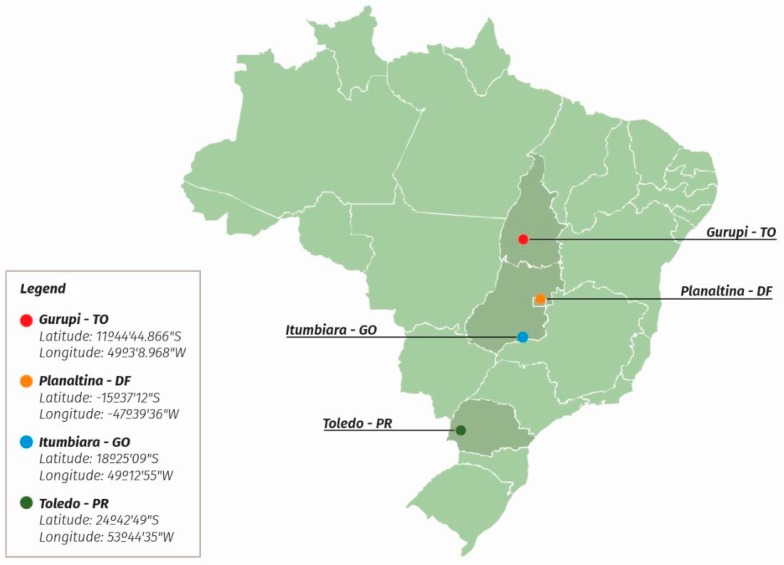
Map of locations and geographic coordinates of the experiments.

**Table 1 toxins-11-00215-t001:** Percentage of maize samples from artificial inoculation (AI) and natural infection (NI) found with the concentration of fumonisins above the tolerance limit (1.5 μg g^−1^ and 1 μg g^−1^) in different Brazilian states.

Tolerance Limits for Fumonisin B1	Gurupi-TO		Itumbiara-GO		Planaltina-DF		Toledo-PR	
	AI	NI	AI	NI	AI	NI	AI	NI
1.5 μg g^−1^ (Cf, Cc, Cm, Flk, Cnj)	100	70	60	90	100	50	70	0
1 μg g^−1^ (Ms, Omp)	100	90	70	90	100	50	80	0

Cf (cornflour), CC (corn cream), Cm (corn meal), Flk (flakes), Cnj (“canjica”), Ms (maize starch), Omp (other maize products).

**Table 2 toxins-11-00215-t002:** Average concentration of fumonisin B1 in maize grains from genotypes that received artificial inoculation (AI) influenced by the average temperature of different Brazilian municipalities.

Values	Gurupi-TO	Itumbiara-GO	Planaltina-DF	Toledo-PR
	AI	AI	AI	AI
Fumonisin levels B1	24.84 μg g^−1^	6.58 μg g^−1^	4.74 μg g^−1^	3.38 μg g^−1^
Temperature	27.43 °C	23.1 °C	22.13 °C	13.64 °C

**Table 3 toxins-11-00215-t003:** Results of the thousand kernel weight (TKW), severities (%) and concentrations of fumonisins (FB1 and FB2) (μg g^−1^) from the 10 commercial maize hybrids planted in Gurupi-TO, Itumbiara-GO, Planaltina-DF and Toledo-PR.

**Gurupi-TO**								
Hybrid	SEV		TKW		Fum. FB1 (μg g^−1^)	Fum. FB2 (μg g^−1^)	Fum. B1 Tes. (μg g^−1^)	Fum. B2 Tes. (μg g^−1^)
	%	SD	(g)	SD				
DKPDKB240PRO2	4.11 ab	0.04	258.56 b	3.26	16.17	4.41	2.74	0.41
P3340YHR	3.44 a	0.08	258.56 b	1.93	5.62	1.16	2.76	0.86
P3630H	3.78 a	0.05	295.22 ab	2.57	86.70	30.45	9.33	3.44
32R48YH	6.56 c	0,08	319.89 ab	2.00	64.80	25.05	16.20	4.20
MFRDKB310PRO2	2.56 a	0.10	338.00 a	2.28	2.69	0.65	0.40	0.13
P3250	4.00 a	0.10	296.33 ab	3.98	32.40	11.43	5.03	1.39
DKPDKB390PRO2	2.78 a	0.05	258.33 b	3.27	12.99	2.89	1.29	0.45
30K75Y	4.11 ab	0.17	283.56 ab	2.47	16.17	4.77	6.45	1.54
P4285H	3.89 a	0.09	296.56 ab	2.09	9.13	3.46	1.15	0.39
DOW30A37PW	4.56 ab	0.13	254.56 b	2.34	1.68	0.49	9.44	5.21
Average	3.98		285.96		24.84	8.48	5.48	1.80
**Itumbiara-GO**								
Hybrid	SEV		TKW		Fum. FB1 (μg g^−1^)	Fum. FB2 (μg g^−1^)	Fum. B1 Tes. (μg g^−1^)	Fum. B2 Tes. (μg g^−1^)
	%	SD	(g)	SD				
DKPDKB240PRO2	6.11 b	0.15	159.89 c	5.53	5.66	1.30	24.84	10.44
P3340YHR	6.22 b	0.13	233.44 ab	3.31	8.04	2.80	1.79	0.35
P3630H	4.67 b	0.06	246.44 ab	7.05	0.71	0.13	6.48	2.52
32R48YH	4.67 ab	0.06	258.67 ab	4.61	0.32	0.00	0.00	0.00
MFRDKB310PRO2	4.56 ab	0.08	272.00 a	4.03	8.67	2.23	3.52	1.20
P3250	5.33 ab	0.11	217.44 abc	3.26	18.81	5.61	37.45	9.10
DKPDKB390PRO2	4.33 a	0.11	230.67 abc	3.59	1.11	0.32	4.41	1.61
30K75Y	6.11 b	0.15	192.33 bc	3.34	12.00	4.11	32.40	8.31
P4285H	2.89 a	0.04	279.89 a	2.07	0.57	0.00	12.78	5.49
DOW30A37PW	5.44 b	0.10	190.11 bc	1.24	9.93	3.32	14.16	5.55
Average	5.03		228.09		6.58	1.98	13.78	4.46
**Planaltina-DF**								
Hybrid	SEV		TKW		Fum. FB1 (μg g^−1^)	Fum. FB2 (μg g^−1^)	Fum. B1 Tes. (μg g^−1^)	Fum. B2 Tes. (μg g^−1^)
	%	SD	(g)	SD				
DKPDKB240PRO2	3.00 a	0.19	285.00 b	2.49	3.11	0.76	1.60	0.52
P3340YHR	3.33 ab	0.18	341.89 ab	5.04	3.29	0.90	2.59	0.96
P3630H	3.22 ab	0.15	312.89 ab	2.45	6.95	2.63	1.59	0.45
32R48YH	5.33 b	0.11	381.78 a	1.88	11.07	3.45	5.58	1.95
MFRDKB310PRO2	2.78 a	0.11	352.33 ab	1.52	5.79	1.67	0.33	0.13
P3250	3.44 ab	0.24	377.44 a	2.32	1.68	0.62	0.31	0.00
DKPDKB390PRO2	3.67 ab	0.15	329.56 ab	2.11	1.79	0.47	0.30	0.16
30K75Y	4.00 ab	0.15	325.44 ab	1.87	4.60	0.99	2.53	0.70
P4285H	2.67 a	0.10	368.67 a	1.27	4.00	0.80	0.65	0.40
DOW30A37PW	3.67 ab	0.15	313.44 ab	1.29	5.13	1.53	0.00	0.00
Average	3.51		338.84		4.74	1.38	1.55	0.53
**Toledo-PR**								
Hybrid	SEV		TKW		Fum. FB1 (μg g^−1^)	Fum. FB2 (μg g^−1^)	Fum. B1 Tes. (μg g^−1^)	Fum. B2 Tes. (μg g^−1^)
	%	SD	(g)	SD				
DKPDKB240PRO2	2.11 a	0.09	302.67 ab	2.20	1.48	0.30	0.00	0.00
P3340YHR	3.33 a	0.15	271.33 ab	5.24	0.00	0.00	0.00	0.00
P3630H	3.00 a	0.09	307.00 ab	4.16	7.57	2.27	0.00	0.00
32R48YH	3.56 a	0.14	240.56 b	4.85	0.20	0.00	0.00	0.00
MFRDKB310PRO2	3.11 a	0.13	288.56 ab	6.72	2.30	0.80	0.27	0.00
P3250	3.78 a	0.20	288.33 ab	4.12	4.01	1.09	0.00	0.00
DKPDKB390PRO2	2.44 a	0.06	267.33 ab	4.34	4.45	2.12	0.27	0.00
30K75Y	2.56 a	0.19	301.00 ab	3.43	2.39	0.47	0.00	0.00
P4285H	2.78 a	0.13	321.22 a	2.83	4.56	1.34	0.00	0.00
DOW30A37PW	2.89 a	0.14	290.89 ab	3.10	6.87	2.02	0.00	0.00
Average	2.96		287.89		33.83	1.04	0.055	0.00

Means followed by the same letter in the column do not differ significantly by a Tukey test at 5% probability. SD: Standard deviation.

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
