# Peer review of "Fumonisin Production by Fusarium verticillioides in Maize Genotypes Cultivated in Different Environments"

_toxins, 2019, doi:10.3390/toxins11040215_

Round 1
Reviewer 1 Report
The manuscript has been modified according to the suggestion and improved. The following Reference is missing and it should be added: Petra Mikušová, Antónia Šrobárová, Michael Sulyok, Antonello Santini Fusarium fungi and associated metabolites presence on grapes from Slovakia. Mycotoxin Research, (2013), 29, 97-102. It can be recommened for publication with this minor revision.
Author Response
Revision Note on Ms. No. Toxins-477688
Reviewer 1:
The manuscript has been modified according to the suggestion and improved. The following Reference is missing and it should be added: Petra Mikušová, Antónia Šrobárová, Michael Sulyok, Antonello Santini Fusarium fungi and associated metabolites presence on grapes from Slovakia. Mycotoxin Research, (2013), 29, 97-102. It can be recommened for publication with this minor revision.
Answer: Thanks for the recommendation. The literature was inserted in the text. (Line 39, 509)
Reviewer 2 Report
Manuscript ID: Toxins_477688
Title: Fumonisin production by Fusarium verticillioides in maize genotypes cultivated in different environments
Authors: Blind review
Recommendation: Major Review
The quality of the language has been improved in this updated version of the manuscript. However, there still are some sentences that are not clearly written or should be improved, and minor errors (see for example, line 27 “reported”, line 33 “conditions”, lines 67-69, line 71 “analyses”, lines 120-121, lines 347-348,…). In addition, species names must be written in italics (see the abstract). There are some references that have not been included in the list, such as Mulé et al (2004). Please, check all references again. By the way, do the authors really extended the annealing in each PCR cycle for 1.5 minutes?
Nevertheless, my major point is, again, the quantitative values given for some parameters in the Tables. In their replies to the comments done to the first version of the manuscript, the authors state that: “There are standard deviation values for disease severity and grain weight for each genotype at the sites where trials were conducted.” But I cannot see these SD values in Table 3 or Figure 2. The authors should include SD values to improve reliability of quantitative data. BTW, in line 318, shouldn´t it be Figure 2 instead of Figure 4? In Table 2, the authors suggest that there is a correlation between the average temperature of each region and fumonisin B1 levels, because its concentration decreases with the average temperature. However, the detailed quantitative data in Table 3, clearly show that this doesn´t hold true for all maize genotypes analyzed (see for example, DOW30A37PW, which in Toledo, with 13 ºC of average temperature, produces more FB1 than in Planaltina, with 22 ºC of average temperature). BTW, the authors should present the hybrids in the same order in each subtable (region) of Table 3. Again, I believe there is no way to make such a biased statement as the one done in Table 2.
Taking everything into consideration, my decision is Major Review.
Author Response
Article: Fumonisin production by Fusarium verticillioides in maize genotypes cultivated in different environments
Revision Note on Ms. No. Toxins-477688
Reviewer 2:
The quality of the language has been improved in this updated version of the manuscript. However, there still are some sentences that are not clearly written or should be improved, and minor errors (see for example, line 27 “reported”, line 33 “conditions”, lines 67-69, line 71 “analyses”, lines 120-121, lines 347-348,…).
Answer: The authors are grateful for the comments. Corrections were made in these parameters.
In addition, species names must be written in italics (see the abstract).
Answer: The corrections were done. See lines 6 and 12.
There are some references that have not been included in the list, such as Mulé et al (2004). Please, check all references again.
Answer: The corrections were done. (Line 596)
By the way, do the authors really extended the annealing in each PCR cycle for 1.5 minutes?
Answer: Sorry for the mistake. The original literature was consulted and the spelling error corrected in the text. The time was correct for 1 minute and 56 °C. (Lines 449, 450)
Nevertheless, my major point is, again, the quantitative values given for some parameters in the Tables. In their replies to the comments done to the first version of the manuscript, the authors state that: “There are standard deviation values for disease severity and grain weight for each genotype at the sites where trials were conducted.” But I cannot see these SD values in Table 3 or Figure 2. The authors should include SD values to improve reliability of quantitative data.
Answer: The standard deviation for severity and thousand kernel weight was inserted in Table 3 as specified by the reviewer.
BTW, in line 318, shouldn´t it be Figure 2 instead of Figure 4?
Answer: The authors agree with the reviewer. There was an error citing Figure 4 instead of Figure 2. The necessary correction was made.
In Table 2, the authors suggest that there is a correlation between the average temperature of each region and fumonisin B1 levels, because its concentration decreases with the average temperature. However, the detailed quantitative data in Table 3, clearly show that this doesn´t hold true for all maize genotypes analyzed (see for example, DOW30A37PW, which in Toledo, with 13 ºC of average temperature, produces more FB1 than in Planaltina, with 22 ºC of average temperature).
Answer: The following text has been added to the paragraph for a better understanding: “However, this occurs considering the average of the cultivars added in each region (Table 2). Observing the amount of fumonisin for each genotype, those that showed a positive correlation with the temperature were 30K75Y, 32R48YH, DKPDKB240PRO2, DKPDKB390PRO2, P3250, P3340YHR and P3630H (Table S2).”
BTW, the authors should present the hybrids in the same order in each subtable (region) of Table 3.
Answer: The corrections were done. Please see table 3.
Again, I believe there is no way to make such a biased statement as the one done in Table 2.
Answer: In the response above, the necessary correction was made specifying which genotypes produced more fumonisin with increasing temperature, taking into account the results of the Pearson correlation presented in Table S2.
Round 2
Reviewer 2 Report
The authors have adressed almost all of my comments; however, as they have done in Table 3, they should also include SD bars in the graphs of Figure 2 showing TKW and severity values for each genotype and in each region.
In addition, please correct "analyzis" in line 73.
After those corrections, the manuscript should be, in my opinion, accepted in Toxins.
Author Response
Article: Fumonisin production by Fusarium verticillioides in maize genotypes cultivated in different environments
Revision Note on Ms. No. Toxins-477688
Cindy Sun
Assistant Editor
Toxins Editorial Office
E-mail: cindy.sun@mdpi.com
Dear Dr Cindy Sun,
In response to the concerns/suggestions raised by the Toxins reviewers that revised our manuscript ( “Fumonisin production by Fusarium verticillioides in maize genotypes cultivated in different environment”), we are very glad to let you know that we took all of them in consideration in the revised version of manuscript. We are more than sure that their contributions significantly increased the scientific quality of our manuscript.
Please find below our answers in a point-by-point format:
Reviewer 2:
The authors have adressed almost all of my comments; however, as they have done in Table 3, they should also include SD bars in the graphs of Figure 2 showing TKW and severity values for each genotype and in each region.
Answer: The authors appreciate the exceptional contribution of the reviewer for the improvement of the article. The standard deviations for grain mass and severity were included in figure 2, as suggested. The values are presented in 5:1 scale, for better viewing, because the original values are small and not representative in the graph.
In addition, please correct "analyzis" in line 73.
Answer: The correction was done.
